# Fast Training of Large Kernel Models with Delayed Projections

**Amirhesam Abedsoltan**
UC San Diego
abedsol1@ucsd.edu

**Siyuan Ma**\*
Google
siyuan@siyuanm.com

**Parthe Pandit**
IIT Bombay
pandit@iitb.ac.in

**Mikhail Belkin**
UC San Diego
mbelkin@ucsd.edu

## Abstract

Classical kernel machines have historically faced significant challenges in scaling to large datasets and model sizes—a key ingredient that has driven the success of neural networks. In this paper, we present a new methodology for building kernel machines that can scale efficiently with both data size and model size. Our algorithm introduces delayed projections to Preconditioned Stochastic Gradient Descent (PSGD) allowing the training of much larger models than was previously feasible. We validate our algorithm, EigenPro 4, across multiple datasets, demonstrating drastic training speedups without compromising the performance. Our implementation is publicly available at: https://github.com/EigenPro/EigenPro.

## 1 Introduction

Kernel methods have strong theoretical foundations and broad applicability. They have also served as the foundation for understanding many significant phenomena in modern machine learning [9, 4, 3, 28]. Despite these advantages, the scalability of kernel methods has remained a persistent challenge, particularly when applied to large datasets. Addressing this limitation is critical for expanding the utility of kernel-based techniques in modern machine learning applications.

A naive approach for training kernel machines is to directly solve the equivalent kernel matrix inversion problem. In general, the computational complexity of this approach is $O(n^3)$, where $n$ is the number of training samples. Thus, computational cost grows rapidly with the size of the dataset, making it computationally intractable for datasets with more than $\sim 10^5$ data points.

To address this challenge, various methods employing iterative algorithms and approximations have been proposed. Among these, Gradient Descent (GD)-based algorithms like Pegasos [21] and EigenPro 1.0,2.0 [13, 14] have significantly reduced the computational complexity to $O(n^2)$. These methods, adaptable for stochastic settings, offer more efficient implementations. Nevertheless, the scalability of kernel machines remains constrained by the inherent linkage between the model size and the training set.

Furthermore, the Nyström methods have emerged as a favored approach for scaling kernel machines, with seminal works with [26] paving the way. Methods such as NYTRO [5], Falkon [20] and ASkotch [19] leverage the Nyström Approximation (NA) in combination with other strategies to enhance performance. NYTRO merges NA with gradient descent to improve condition number, ASkotch combines it with block coordinate descent, whereas Falkon combines it with the Conjugate Gradient method, facilitating the handling of large training sets. However, these strategies are limited by model size due to memory restrictions, exhibiting quadratic scaling in relation to the size of the

---

\*Work completed during author's time at Google. Current affiliation: Lynko.

39th Conference on Neural Information Processing Systems (NeurIPS 2025).

| Algorithm | FLOPS | | Memory |
| | setup | per batch | |
|---|---|---|---|
| **EigenPro 4** | $O(1)$ | $O(p)$ | $O(p)$ |
| EigenPro 3 | $O(1)$ | $O(p^2)$ | $O(p)$ |
| Falkon | $O(p^3)$ | $O(p)$ | $O(p^2)$ |

Figure 1: Comparison of per-epoch time complexity across solvers as a function of model size $p$. Performance in terms of classification test accuracy (indicated as percentages) is annotated next to each data point, showing that EP4 maintains superior or comparable performance across all model sizes. Details of the experiment and hardware can be found in Appendix C.

model. For instance, scaling Falkon [16] method to a model size of $512,000$ necessitates over 1TB of RAM, surpassing the capacity of most high-end servers available today.

Other lines of work in the Gaussian Processes literature, e.g., [22, 27, 7, 15], use so-called *inducing points* to control model complexity. However, these methods face similar scaling issues as they require quadratic memory in terms of the number of inducing points, preventing large models.

Recently, EigenPro 3.0 was introduced in [1]. Unlike previous versions, EigenPro 3.0 distangles the model from the training set, similar to Falkon, but with the added advantage that its memory requirements scales linearly with the model size. This advancement makes it feasible to tackle kernel models of sizes previously deemed unattainable. However, its per iteration time complexity remains quadratic relative to the model size, significantly slowing its practical application.

In this paper, we build upon EigenPro 3.0 and introduce EigenPro 4.0. This new algorithm retains the advantageous features of EigenPro 3.0, such as decoupling the model from the training set and linear scaling in memory complexity. Moreover, it significantly improves upon the time complexity, achieving amortized linear scaling per iteration with respect to model size. Empirically we observe that the proposed algorithm converges in fewer epochs, without compromising generalization performance.

## 1.1 Main contribution

Our method for kernel machine problems achieves three key advantages: (1) linear amortized time complexity per iteration, (2) linear memory scaling with model size, (3) comparable or superior performance compare to existing methods while demonstrating up to 600× speedup in our experiments. Figure 1 demonstrates these benefits on the CIFAR5M data set.

## 1.2 Organization of the Paper

The remainder of this paper is organized as follows. Section 2 provides the necessary background and preliminaries. In Section 3, we derive the complete algorithm, and introduce the key insights and techniques that enable its dramatic improvement in computational efficiency. Finally, Section 4 presents extensive experimental results across multiple datasets and model sizes.

## 2 Notation and Background

In what follows, functions are lowercase letters $a$, sets are uppercase letters $A$, vectors are lowercase bold letters $\boldsymbol{a}$, matrices are uppercase bold letters $\boldsymbol{A}$, operators are calligraphic letters $\mathcal{A}$, spaces and sub-spaces are boldface calligraphic letters $\boldsymbol{\mathcal{A}}$.

**General kernel models.** Following EigenPro 3.0 [1] notations, given training data $(X, \boldsymbol{y}) = \left\{\boldsymbol{x}_i \in \mathbb{R}^d, y_i \in \mathbb{R}\right\}_{i=1}^n$, *General kernel models* are models of the form,

$$f(\boldsymbol{x}) = \sum_{i=1}^p \alpha_i K(\boldsymbol{x}, \mathbf{z}_i).$$

Here, $K : \mathbb{R}^d \times \mathbb{R}^d \to \mathbb{R}$ is a positive semi-definite symmetric kernel function and $Z = \{\mathbf{z}_i \in \mathbb{R}^d\}_{i=1}^p$ are *centers*, which are not necessarily in the training set $X$. We will refer to $p$ as the *model size*. We define $\mathcal{H}$ as the unique reproducing kernel Hilbert space (RKHS) corresponding to $K$.

**Loss function.** Our goal will be to find the solution to the following infinite-dimensional Mean Squared Error (MSE) problem for general kernel models,

$$\underset{f \in \mathcal{H}}{\text{minimize}} \ L(f) = \frac{1}{2} \sum_{i=1}^n (f(\boldsymbol{x}_i) - y_i)^2, \qquad \text{subject to} \quad f \in \boldsymbol{\mathcal{Z}} := \text{span}\Big(\{K(\cdot, \mathbf{z}_j)\}_{j=1}^p\Big). \quad (1)$$

**Evaluations and kernel matrices.** The vector of evaluations of a function $f$ over a set $X = \{\boldsymbol{x}_i\}_{i=1}^n$ is denoted $f(X) := (f(\boldsymbol{x}_i)) \in \mathbb{R}^n$. For sets $X$ and $Z$, with $|X| = n$ and $|Z| = p$, we denote the kernel matrix $K(X, Z) \in \mathbb{R}^{n \times p}$, while $K(Z, X) = K(X, Z)^\top$. Similarly, $K(\cdot, X) \in \mathcal{H}^n$ is a vector of $n$ functions, and we use $K(\cdot, X)\boldsymbol{\alpha} := \sum_{i=1}^n K(\cdot, \boldsymbol{x}_i)\alpha_i \in \mathcal{H}$, to denote their linear combination. Finally, for an operator $\mathcal{A}$, a function $a$, and a set $A = \{\boldsymbol{a}_i\}_{i=1}^k$, we denote the vector of evaluations,

$$\mathcal{A}\{a\}(A) := (b(\boldsymbol{a}_i)) \in \mathbb{R}^k \qquad \text{where} \quad b = \mathcal{A}(a). \quad (2)$$

**Fréchet derivative.** Given a function $J : \mathcal{H} \to \mathbb{R}$, the Fréchet derivative of $J$ with respect to $f$ is a linear functional, denoted $\nabla_f J$, such that for $h \in \mathcal{H}$

$$\lim_{\|h\|_{\mathcal{H}} \to 0} \frac{|J(f + h) - J(f) - \nabla_f J(h)|}{\|h\|_{\mathcal{H}}} = 0. \quad (3)$$

Since $\nabla_f J$ is a linear functional, it lies in the dual space $\mathcal{H}^*$. Since $\mathcal{H}$ is a Hilbert space, it is self-dual, whereby $\mathcal{H}^* = \mathcal{H}$. If $f$ is a general kernel model, and $L$ is the square loss for a given dataset $(X, \boldsymbol{y})$, i.e., $L(f) := \frac{1}{2} \sum_{i=1}^n (f(\boldsymbol{x}_i) - y_i)^2$ we can apply the chain rule, and using reproducing property of $\mathcal{H}$, and the fact that $\nabla_f \langle f, g \rangle_{\mathcal{H}} = g$, we get, that the Fréchet derivative of $L$, at $f = f_0$ is,

$$\nabla_f L(f_0) = \sum_{i=1}^n (f_0(\boldsymbol{x}_i) - y_i)\nabla_f f(\boldsymbol{x}_i) = K(\cdot, X)(f_0(X) - \boldsymbol{y}). \quad (4)$$

**Hessian operator.** The Hessian operator $\nabla_f^2 L : \mathcal{H} \to \mathcal{H}$ for the square loss is given by

$$\mathcal{K} := \sum_{i=1}^n K(\cdot, \boldsymbol{x}_i) \otimes K(\cdot, \boldsymbol{x}_i), \qquad \mathcal{K}\{f\}(\mathbf{z}) = \sum_{i=1}^n K(\mathbf{z}, \boldsymbol{x}_i)f(\boldsymbol{x}_i) = K(\mathbf{z}, X)f(X), \quad (5)$$

where $\otimes$ denotes the *outer product* between functions in the RKHS, defined as $(a \otimes b)(\cdot) = a(\cdot) \langle b, \cdot \rangle_{\mathcal{H}}$. The operator $\mathcal{K}$ has non-negative eigenvalues, which we order as $\lambda_1 \geq \lambda_2 \geq \cdots \geq \lambda_n \geq 0$. Hence, we can write its eigen-decomposition as $\mathcal{K} = \sum_{i=1}^n \lambda_i \, \psi_i \otimes \psi_i$. Combining Equations 4 and 5, we can rewrite the Fréchet derivative of the loss function as

$$\nabla_f L(f_0)(z) = \mathcal{K}\{f_0(X) - \boldsymbol{y}\}(z). \quad (6)$$

**Exact minimum norm solution.** The closed-form minimum $\|\cdot\|_{\mathcal{H}}$ norm solution to the problem defined in equation (1) is given by:

$$\hat{f} := K(\cdot, Z)K^\dagger(Z, X)\boldsymbol{y}, \quad (7)$$

where $\dagger$ is the pseudoinverse or Moore–Penrose inverse. In the case of $X = Z$ it simplifies to $\hat{f} := K(\cdot, X)K^{-1}(X, X)\boldsymbol{y}$.

**Gradient Descent (GD).** If we apply GD on the optimization problem in 1, with learning rate $\eta$, in the $\mathcal{H}$ functional space, the update is as following,

$$f_{t+1} = f_t - \eta \cdot \nabla_f L(f_t) = f_t - \eta K(\cdot, X)(f_t(X) - \boldsymbol{y}). \quad (8)$$

The first point to note is that the derivative lies in $\boldsymbol{\mathcal{X}} := \text{span}\Big(\{K(\cdot, \boldsymbol{x}_j)\}_{j=1}^n\Big)$ rather than in $\boldsymbol{\mathcal{Z}}$. Therefore, when $X \neq Z$, SGD cannot be applied in this form. We will revisit this issue later. The second point is that in the case of $X = Z$, traditional *kernel regression* problem, the convergence of SGD depends on the condition number of $\mathcal{K}$. Simply put, this is the ratio of the largest to the smallest

non-zero singular value of the Hessian operator defined in 5. It is known that for general kernel models this condition number is ill conditioned and converges slow, see [2] for more details on this.

**EigenPro .** Prior work, EigenPro , by [13], addresses the slow convergence of SGD by introducing a preconditioned stochastic gradient descent mechanism in Hilbert spaces. The update rule is the same as Equation 9 but with an additional preconditioner $\mathcal{P} : \mathcal{H} \to \mathcal{H}$ applied to the gradient,

$$f_{t+1} = f_t - \eta \cdot \mathcal{P} \nabla_f L(f_t). \tag{9}$$

In short, the role of the preconditioner $\mathcal{P}$ is to suppress the top eigenvalues of the Hessian operator $\mathcal{K}$ to improve the condition number. We next explicitly define $\mathcal{P}$.

**Definition 1** (Top-$q$ Eigensystem). Let $\lambda_1 > \lambda_2 > \ldots > \lambda_n$ be the eigenvalues of a Hermitian matrix $\boldsymbol{A} \in \mathbb{R}^{n \times n}$, where for standard unit vector $\boldsymbol{e}_i$, we have $\boldsymbol{A}\boldsymbol{e}_i = \lambda_i \boldsymbol{e}_i$. We define the tuple $(\Lambda_q, \boldsymbol{E}_q, \lambda_{q+1})$ as the top-$q$ eigensystem, where:

$$\Lambda_q := \operatorname{diag}(\lambda_1, \lambda_2, \ldots, \lambda_q) \in \mathbb{R}^{q \times q}, \qquad \boldsymbol{E}_q := [\boldsymbol{e}_1, \boldsymbol{e}_2, \ldots, \boldsymbol{e}_q] \in \mathbb{R}^{n \times q}. \tag{10}$$

**Preconditioner.** Using Definition 1, let $(\Lambda_q, \boldsymbol{E}_q, \lambda_{q+1})$ be the top-$q$ eigensystem of $K(X, X)$, the preconditioner $\mathcal{P} : \mathcal{H} \to \mathcal{H}$ can be explicitly written as, $\mathcal{P} := \mathcal{I} - \sum_{i=1}^{q} \left( 1 - \frac{\lambda_{q+1}}{\lambda_q} \right) \psi_i \otimes \psi_i$.

**Nyström approximate preconditioner.** EigenPro 2, introduced by [14], implements a stochastic approximation for $\mathcal{P}$ based on the Nyström extension, thereby reducing the time and memory complexity compared to EigenPro . The first step is to approximate the Hessian operator using the Nyström extension as follows,

$$\mathcal{K}^s := \sum_{k=1}^{s} K(\cdot, \boldsymbol{x}_{i_k}) \otimes K(\cdot, \boldsymbol{x}_{i_k}) = \sum_{i=1}^{s} \lambda_i^s \cdot \psi_i^s \otimes \psi_i^s. \tag{11}$$

This is a Nyström approximation of $\mathcal{K}$ using $s$ uniformly random samples from $X$, referred to as $X_s$, where $(\Lambda_q^s, \boldsymbol{E}_q^s, \lambda_{q+1}^s)$ represents the corresponding top-$q$ eigensystem of $K(X_s, X_s)$. Using this approximation, we can define the approximated preconditioner as follows, $\mathcal{P}^s := \mathcal{I} - \sum_{i=1}^{q} \left( 1 - \frac{\lambda_{q+1}^s}{\lambda_q^s} \right) \psi_i^s \otimes \psi_i^s$. For more details on the performance of this preconditioner compared to the case of $s = n$, see [2], who showed that choosing $s \gtrsim \log^4 n$ is sufficient.

Of particular importance is the action of this preconditioner on any function of the form $K(\cdot, A)\boldsymbol{u}$.

$$\mathcal{P}^s K(\cdot, A)\boldsymbol{u} = K(\cdot, A)\boldsymbol{u} - \sum_{i=1}^{q} \left( 1 - \frac{\lambda_{q+1}^s}{\lambda_q^s} \right) \psi_i \psi_i(A)^\top \boldsymbol{u} \tag{12a}$$

$$= K(\cdot, A)\boldsymbol{u} - \sum_{i=1}^{q} \left( 1 - \frac{\lambda_{q+1}^s}{\lambda_q^s} \right) \frac{K(\cdot, X_s)\boldsymbol{e}_i}{\sqrt{\lambda_i}} \frac{\boldsymbol{e}_i^\top K(X_s, A)}{\sqrt{\lambda_i}} \boldsymbol{u} \tag{12b}$$

$$= K(\cdot, A)\boldsymbol{u} - K(\cdot, X_s)\boldsymbol{E}_q^s \boldsymbol{D}_q \boldsymbol{E}_q^{s^\top} K(X_s, A)\boldsymbol{u} \tag{12c}$$

where $\boldsymbol{D}_q := \Lambda_q^{-1} - \lambda_{q+1}\Lambda_q^{-2}$.

**EigenPro 3.** The primary limitation of EigenPro 2 was its inability to handle cases where $Z \neq X$, a necessary condition for disentangling the model and the training set. EigenPro 3 overcomes this limitation by recognizing that although the gradients in (4) may not lie within $\boldsymbol{\mathcal{Z}}$, it is possible to project them back to $\boldsymbol{\mathcal{Z}}$. Consequently, EigenPro 3 can be summarized as follows:

$$f_{t+1} = \operatorname{proj}_{\boldsymbol{\mathcal{Z}}} \left( f_t - \eta \mathcal{P}^s \{ \widetilde{\nabla}_f L(f_t) \} \right), \tag{13}$$

where $\operatorname{proj}_{\boldsymbol{\mathcal{Z}}}(u) := \underset{f \in \boldsymbol{\mathcal{Z}}}{\operatorname{argmin}} \|u - f\|_{\mathcal{H}}^2$ for any $u \in \mathcal{H}$. As shown in [1, Section 4.2 ], the exact projection is, $\operatorname{proj}_{\boldsymbol{\mathcal{Z}}}(u) = K(\cdot, Z)K^{-1}(Z, Z)u(Z)$.

This projection can be interpreted as solving a kernel in $\boldsymbol{\mathcal{Z}}$ and can be approximated using EigenPro 2, as done in [1], with a time complexity that scales quadratically with model size. However, since this projection must be performed after each stochastic step, it becomes the most computationally expensive part of the EigenPro 3 algorithm.

# 3 EigenPro 4: Algorithm Design, Derivation, and Optimization

In this section, we provide a high-level overview and illustrations to highlight the key components of EigenPro 4 and how it significantly reduces training time. We present the EigenPro 4 algorithm in three parts. First, we introduce the algorithm's main components: the pre-projection and projection steps. Then, we detail each of these steps in the following two subsections. Finally, we describe a computational optimization that reduces the runtime of EigenPro 4 by half.

**Scaling Challenge: High Projection Overhead.** The key development of EigenPro 3 over its contemporaries was that it could train general kernel models in $O(p)$ memory. This was a huge improvement over the prior methods which required $O(p^2)$ memory [20, 19]. However, EigenPro 3 has a high cost $O(mp + p^2)$ per batch of data processed, as summarized in the table in Figure 1. This is especially expensive when $m \ll p$, i.e., when the batch size $m$ is small compared to the model size $p$.

**Main Idea: Delayed Projection.** To address computational complexity challenges, EP4 amortizes projection costs by delaying them for $T$ iterations. An effective method for selecting $T$, along with an illustration of the delayed projection mechanism for $T = 4$ (Figure 4), is provided in Appendix B.

In fact, EigenPro 3 can be viewed as a special case of EigenPro 4 when the parameter $T$ is set to 1. Figure 5 in Appendix B illustrates this relationship. Furthermore, as shown in equation (26) in the same appendix, the total training time is minimized when $T$ is proportional to $\frac{p}{m}$, where $m$ denotes the mini-batch size used in SGD. Under this setting, the per-batch training cost becomes $O(p)$.

## 3.1 Derivation of the EigenPro 4 Algorithm

As mentioned previously $T$ is a crucial hyperparameter that determines the frequency of projection back to $\mathcal{Z}$ after every $T$ steps. Before the projection step $T$, at every step when a new batch $(X_m, y_m)$ is fetched, it is added to a set defined as the "temporary centers" set, denoted by $Z_{\mathsf{tmp}}$. Starting with an empty set, $Z_{\mathsf{tmp}} = \emptyset$, we continuously add temporary centers to $Z_{\mathsf{tmp}}$ until the step count reaches $T$.

Formally, the prediction function prior to the projection is no longer fixed and is now expanding. The model can be described as follows:

$$f(x) = \overbrace{\sum_{\mathbf{z} \in Z} \alpha_{\mathbf{z}} K(\boldsymbol{x}, \mathbf{z})}^{\text{original model}} + \overbrace{\sum_{\mathbf{z} \in Z_{\mathsf{tmp}}} \beta_{\mathbf{z}} K(\boldsymbol{x}, \mathbf{z})}^{\text{temporary centers}} \tag{14}$$

where $\alpha_{\mathbf{z}}$ refers to the weights corresponding to the original model center $\mathbf{z}$ and $\beta_{\mathbf{z}}$ refers to the weights corresponding to the temporary model centers. The full EigenPro 4 algorithm has been illustrated in Figure 2 and mathematically can be summarized as following,

$$f_t = \begin{cases} \mathrm{proj}_{\boldsymbol{\mathcal{Z}}}\left(f_{t-1} - \eta \mathcal{P}^s\{\widetilde{\nabla}_f L(f_{t-1})\}\right), & t \equiv 0 \mod T, \\ f_{t-1} - \eta \mathcal{P}^s\{\widetilde{\nabla}_f L(f_t)\}, & \text{otherwise.} \end{cases} \tag{15}$$

where $\widetilde{\nabla} L$ is a stochastic gradient of the loss function computed over a mini-batch of data, and $\mathcal{P}^s$ is the preconditioner.

## 3.2 Steps before the Projection Step: Update for $t < T$

Based on Equation (15), suppose $(X_1, y_1), \ldots, (X_T, y_T)$ are the minibatches of size $m$, and the initial model is $f_0 = K(\cdot, Z)\boldsymbol{\alpha}$. After $t < T$ step, the following holds,

$$f_t = f_{t-1} - \eta \mathcal{P}^s K(\cdot, X_t)(f_{t-1}(X_t) - y_t) = f_0 - \eta \mathcal{P}^s \sum_{i=1}^{t} K(\cdot, X_i)\boldsymbol{g}_i$$

$$\boldsymbol{g}_i := f_{i-1}(X_i) - y_i$$

Replacing $\mathcal{P}^s$ using equation (11), setting $f_0 = K(\cdot, Z)\boldsymbol{\alpha}$, and following the notation introduced in section 2, let $(\Lambda_q, \boldsymbol{E}_q^s, \lambda_{q+1})$ denote the top-$q$ eigensystem of $K(X_s, X_s)$, where $\boldsymbol{E}_q^s \in \mathbb{R}^{s \times q}$ and $\boldsymbol{D}_q := \Lambda_q^{-1} - \lambda_{q+1}\Lambda_q^{-2}$. Then, we can simplify the update above as follows:

$$f_t = K(\cdot, Z)\boldsymbol{\alpha}_0 - \eta \sum_{i=1}^{t} \left( K(\cdot, X_i) - K(\cdot, X_s)\boldsymbol{E}_q^s \boldsymbol{D}_q \boldsymbol{E}_q^{s\top} K(X_s, X_i) \right) \boldsymbol{g}_i \tag{16}$$

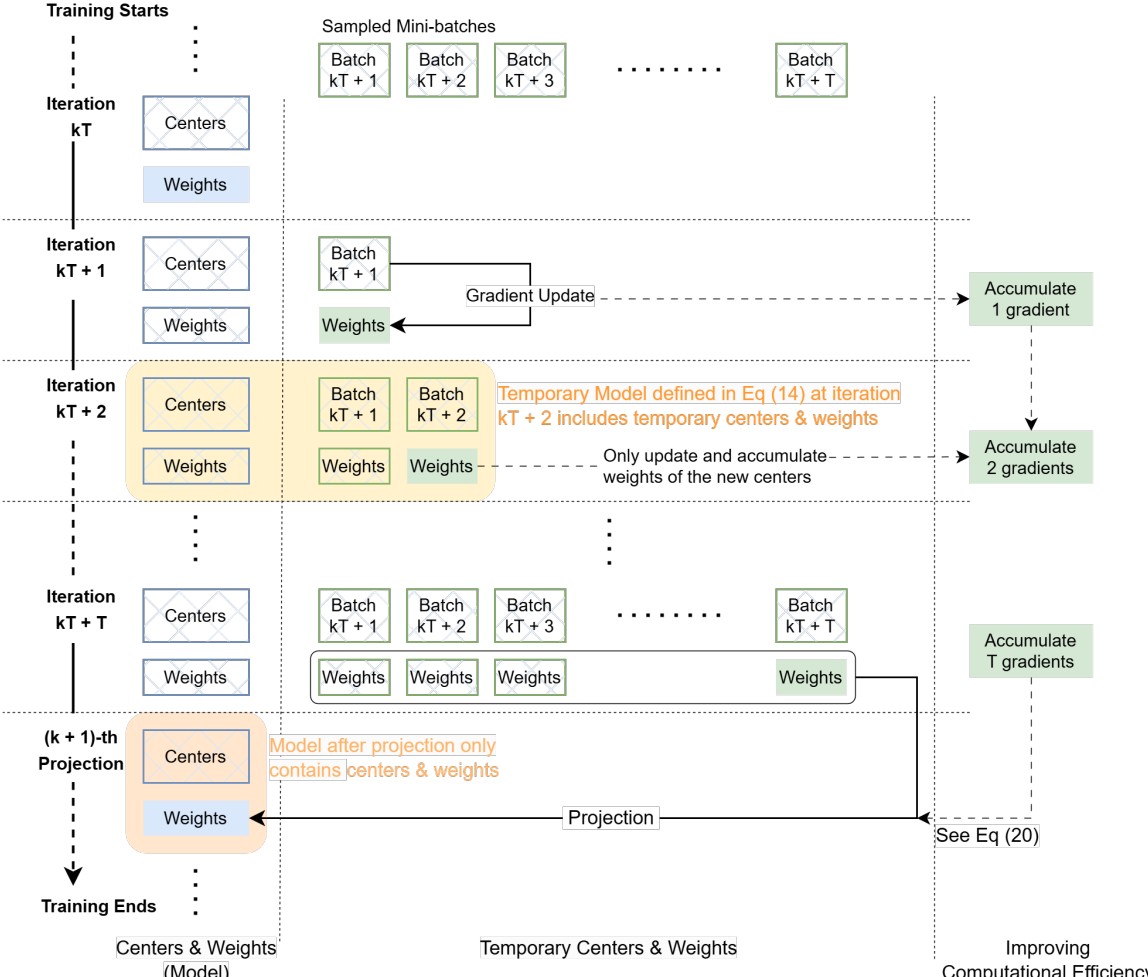

Figure 2: **Schematic of the iterations in EigenPro 4.** The figure illustrates how the model updates are performed over multiple iterations in EigenPro4. Weights are updated using batches, and gradients are accumulated iteratively until a projection step is executed. This approach reduces the computational cost by accumulating gradients before performing the projection, leading to more efficient batch processing.

This update rule implies that after $t$ steps, the weights corresponding to the original centers $Z$ remain unchanged, the weights for the temporary centers $X_i$ are set once to $-\eta \boldsymbol{g}_i$ after they are added, and do not change thereafter, and finally, the weights associated with the Nyström samples $X_s$ are $\eta \sum_{i=1}^{t} \boldsymbol{E}_q^s \boldsymbol{D}_q \boldsymbol{E}_q^{s^\top} K(X_s, X_i) \boldsymbol{g}_i$ which can be updated after each batch via an additive update. This is how we update the weights before projection at step $T$.

### 3.3 Projection Step: Update for $t = T$

Once, we reach step $T$ we need to project $f_T$ into $\boldsymbol{\mathcal{Z}}$, or formally

$$f_T = \text{proj}_{\boldsymbol{\mathcal{Z}}} \left( f_{T-1} - \eta \mathcal{P}^s \{ \widetilde{\nabla}_f L(f_{T-1}) \} \right) \tag{17}$$

$$= \text{proj}_{\boldsymbol{\mathcal{Z}}} \left( f_{T-1} - \eta \left( K(\cdot, X_T) \boldsymbol{g}_T - \eta K(\cdot, X_s) \boldsymbol{E}_q^s \boldsymbol{D} \boldsymbol{E}_q^{s^\top} K(X_s, X_T) \boldsymbol{g}_T \right) \right),$$

Applying Proposition 2 from [1], the solution to this projection problem is as follows,

$$f_T = K(\cdot, Z) K^{-1}(Z, Z) \left( f_{T-1}(Z) - \eta K(Z, X_T) \boldsymbol{g}_T + \eta K(Z, X_s) \boldsymbol{E}_q^s \boldsymbol{D} \boldsymbol{E}_q^{s^\top} K(X_s, X_T) \boldsymbol{g}_T \right) \tag{18}$$

---

**Algorithm 1** EigenPro 4

---

**Require:** Data $(X, \boldsymbol{y})$, centers $Z$, batch size $m$, Nyström size $s$, preconditioner level $q$, projection period $T$

1: Fetch subsample $X_s \subseteq X$ of size $s$
2: $(\Lambda_q, \boldsymbol{E}_q^s, \lambda_{q+1}) \leftarrow$ top-$q$ eigensystem of $K(X_s, X_s)$   and define   $\boldsymbol{D}_q := (\Lambda_q^{-1} - \lambda_{q+1}\Lambda_q^{-2}) \in \mathbb{R}^{q \times q}$
3: **while** Stopping criterion is not reached **do**
4:     $Z_{\mathsf{tmp}} \leftarrow \emptyset, \boldsymbol{\alpha}_{\mathsf{tmp}} \leftarrow \emptyset, \boldsymbol{\alpha}_s \leftarrow \boldsymbol{0}_s, \boldsymbol{h} \leftarrow \boldsymbol{0}_p$
5:     **for** $t = \{1, 2, \ldots, T\}$ **do**
6:         Fetch minibatch $(X_m, \boldsymbol{y}_m)$ of $m$ samples
7:         $\boldsymbol{g}_m \leftarrow K(X_m, Z)\boldsymbol{\alpha} + K(X_m, Z_{\mathsf{tmp}})\boldsymbol{\alpha}_{\mathsf{tmp}} + K(X_m, X_s)\boldsymbol{\alpha}_s - \boldsymbol{y}_m$
8:         $Z_{\mathsf{tmp}}.\text{append}(X_m)$   and   $\boldsymbol{\alpha}_{\mathsf{tmp}}.\text{append}(-\eta \cdot \boldsymbol{g}_m)$
9:         $\boldsymbol{\alpha}_s = \boldsymbol{\alpha}_s + \eta \cdot \boldsymbol{E}_q^s \boldsymbol{D} \boldsymbol{E}_q^{s\top} K(X_s, X_m)\boldsymbol{g}_m$
10:        $\boldsymbol{h} \leftarrow \boldsymbol{h} + K(Z, X_m)\boldsymbol{g}_m - K(Z, X_s)\boldsymbol{E}_q^s \boldsymbol{D} \boldsymbol{E}_q^{s\top} K(X_s, X_m)\boldsymbol{g}_m$
11:    **end for**
12:    $\boldsymbol{\alpha} \leftarrow \boldsymbol{\alpha} - \eta \cdot \text{proj}_{\boldsymbol{Z}}(\boldsymbol{h})$          {Approximate projection implemented using EigenPro 2}
13: **end while**

---

### 3.4   Improving Computational Efficiency

Upon careful examination of the derivations in (16), we observe that $f_{T-1}(Z)$ have already been computed previously. This allows us to efficiently reuse $f_{T-1}(Z)$ as follows,

$$f_{T-1}(Z) = K(Z, Z)\boldsymbol{\alpha}_0 - \eta \left( \sum_{i=1}^{T-1} K(Z, X_i)\boldsymbol{g}_i - K(Z, X_s)\boldsymbol{E}_q^s \boldsymbol{D} \boldsymbol{E}_q^{s\top} K(X_s, X_i)\boldsymbol{g}_i \right) \quad (19)$$

plugging this in (18) we obtain,

$$f_T = K(\cdot, Z)\left(\boldsymbol{\alpha}_0 - \eta K^{-1}(Z, Z)\boldsymbol{h}\right), \quad (20a)$$

$$\boldsymbol{h} := \sum_{i=1}^{T} K(Z, X_i)\boldsymbol{g}_i - K(Z, X_s)\boldsymbol{E}_q^s \boldsymbol{D} \boldsymbol{E}_q^{s\top} \sum_{i=1}^{T} K(X_s, X_i)\boldsymbol{g}_i \quad (20b)$$

### 3.5   Final algorithm

The final EigenPro 4 can be found in Algorithm 1. Note that we follow the same *inexact projection* scheme used in [1] to approximate the exact projection in the last step of the algorithm.

The benefit of this approximation is that we don't need to solve the problem exactly in $\boldsymbol{\mathcal{X}}$, nor do we need to project back to $\boldsymbol{\mathcal{Z}}$ after each iteration. This approach offers the best of both worlds. In the next section, we demonstrate the effectiveness of this approach compared to prior state-of-the-art methods.

### 3.6   Convergence Analysis

In Appendix A, we provide a convergence analysis for the special case of Algorithm 1 where $T = \infty$ and exact projection is used in the projection step. However, a more rigorous theoretical analysis that accounts for the approximations introduced for scability—namely, finite $T$ and inexact projections—is left for future work and is beyond the scope of this paper.

## 4   Numerical experiments

In this section, we demonstrate that our approach achieves orders-of-magnitude speedups over state-of-the-art kernel methods while maintaining comparable or superior generalization performance. We evaluate several kernel methods on the following datasets: (1) CIFAR5M, (2) CIFAR5M[2] [17], (3) ImageNet[1] [6], (4) WebVision[2] [12], and (5) LibriSpeech [18]. Dataset details are provided in Appendix C.

---

[2]Feature extraction using MobileNetV2
[2]Feature extraction using ResNet-18

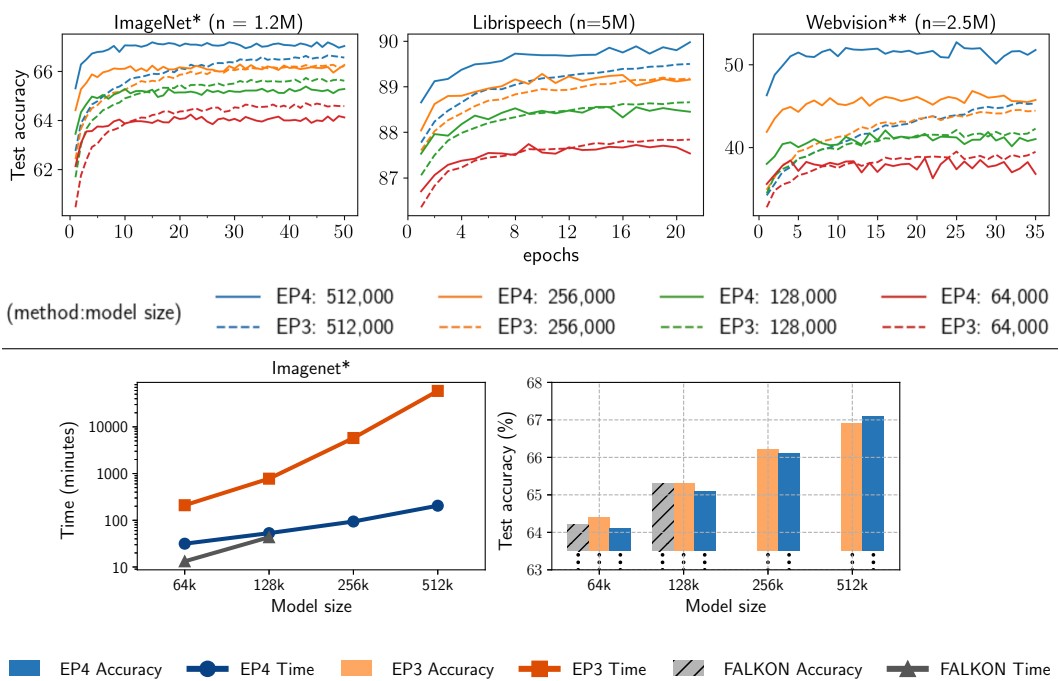

Figure 3: (top) Multi-epoch performance, convergence comparison for EigenPro 3 and EigenPro 4. (bottom) Time and performance comparison for Falkon, EigenPro 3 and EigenPro 4 for ImageNet. Details of the experiments and hardware can be found in Appendix C. (Here $n$ refers to total number of training samples.)

While our method is compatible with any kernel function, we primarily use the Laplace kernel due to its simplicity and strong empirical performance. For completeness, we also report results using the Gaussian and NTK kernels in Appendix C.1.

For multi-class classification, we adopt a one-vs-all decomposition strategy, training a separate binary regressor for each class with targets in $0, 1$. The final prediction is obtained by selecting the class corresponding to the highest predicted value across all binary regressors.

**Substantial Reduction in Per-Epoch Training Time.** EigenPro 4 has substantially reduced the per-epoch training time, making it the most efficient kernel method on modern machine learning hardware. In contrast to performing projection every mini-batch iteration as in EigenPro 3, EigenPro 4 schedules one projection every few iterations such that its amortized cost is comparable to that of the standard iterations. This results in an ideal per-sample complexity $O(p)$, a remarkable improvement over the $O(p^2)$ complexity from EigenPro 3.

In Table 1, we evaluate the performance and computational timing for a single epoch of our proposed model against established kernel regression methods. As noted earlier, Falkon exhibits limitations due to its quadratic memory complexity. For the CIFAR5M* dataset, training with a model size of 512,000 required 1.3TB of RAM, while scaling to 1M model size necessitated over 5TB. Resource constraints limited our Falkon benchmarks to model sizes of 128,000 and 64,000 for the remaining datasets. While EigenPro 3 addresses these memory constraints, it demonstrates significant computational overhead, particularly evident in the Librispeech dataset where our method, EigenPro 4, achieves a $411\times$ speedup. Notably, EigenPro 4 maintains comparable or superior performance across all evaluated datasets relative to both baseline methods.

**Linear Scaling with Model Size.** The total training time and memory usage of EigenPro 4 scales linearly with the model size. In comparison, the time required for a single EigenPro 3 iteration grows quadratically with the model size, while the preprocessing time for Falkon grows cubically. Furthermore, the memory demand of Falkon increases quadratically with the model size. In practice, we are unable to run it with large model sizes, e.g., 128,000 centers for ImageNet data.

| Model size | Method | CIFAR5M*($n = 5M$) | CIFAR5M ($n = 6M$) | Librispeech ($n = 10M$) | Webvision ($n = 5.2M$) |
|---|---|---|---|---|---|
| | EigenPro 4 | 5m (4.6x, 88%) | 3m (**15x, 69%**) | 16m (9.1x, **86.8%**) | 2m (**45.5x, 24.3%**) |
| p = 64K | EigenPro 3 | 23m (1x, **88.3%**) | 45m (1x, 68.8%) | 145m (1x, 85.4%) | 91m (1x, 24%) |
| | Falkon | 3m (**7.67x**, 86.1%) | 5m (9x, 57.7%) | 9m (**16.11x**, 81.0%) | 4m (22.75x, 21.7%) |
| | EigenPro 4 | 5m (**10x**, 88.25%) | 4m (**26.25x, 70.9%**) | 19m (**17.95x, 87.8%**) | 4m (**49.75x, 24.9%**) |
| p = 128K | EigenPro 3 | 50m (1x, **88.42%**) | 105m (1x, 70.3%) | 341m (1x, 84.75%) | 199m (1x, 24.5%) |
| | Falkon | 9m (5.56x, 86.55%) | 11m (9.55x, 59.4%) | 21m (16.24x, 82.30%) | 13m (15.31x, 22.4%) |
| | EigenPro 4 | 7m (**18.3x, 88.61%**) | 6m (**130.8x, 71.8%**) | 24m (**120x, 88.33%**) | 5m (**106.2x, 26%**) |
| p = 256K | EigenPro 3 | 128m (1x, **88.61%**) | 785m (1x, 70.53%) | $\approx$ 2 days (1x) | 531m (1x, 25.52%) |
| | Falkon | 38m (3.37x, 86.73%) | OOM | OOM | OOM |
| | EigenPro 4 | 12m (**44.25x, 88.58%**) | 10m (> **288x**, 72.9%) | 36m (> **200x, 88.89%**) | 11m (**240x, 27.3%**) |
| p = 512K | EigenPro 3 | 531m (1x, 88.56%) | > 2 days (1x) | > 5 days (1x) | 2 days (1x) |
| | Falkon | 240m (2.21x, 86.71%) | OOM | OOM | OOM |
| | EigenPro 4 | 21m (> **274x, 88.7%**) | 17m (> **508x, 73.8%**) | 70m (> **411x, 89.5%**) | 21m (> **686x, 29.3%**) |
| p = 1M | EigenPro 3 | > 4 days (1x) | > 6 days (1x) | >20 days (1x) | > 10 days (1x) |
| | Falkon | OOM | OOM | OOM | OOM |

Table 1: Runtime (in minutes) comparison of EigenPro 4, EigenPro 3, and Falkon across different model sizes and datasets after 1 epoch. The values in parentheses represent the speedup over EigenPro 3 and the final accuracy. OOM indicates an Out-of-Memory error. Details of the experiments and hardware can be found in Appendix C.

We summarize all empirical results in Figure 3 and demonstrate that our method achieves both linear memory complexity and linear time complexity (empirically verified) with respect to model size, offering the best of both worlds. For the ImageNet dataset, we trained all methods until convergence. While EigenPro 3 does not have the quadratic memory scaling problem, Figure 3 shows that even for a relatively small dataset like ImageNet with 1M data points, training a model size of 512,000 centers requires approximately 43 days on a single GPU to reach convergence (about 100 epochs). In contrast, our proposed model achieves convergence in approximately 3 hours, requiring only 15 epochs, with each epoch being significantly more efficient than EigenPro 3 (see Table 1).

**Faster Convergence with EigenPro 4.** EigenPro 4 generally demonstrates the fastest convergence among all tested methods. In certain cases, such as ImageNet with 1.2 million model centers, EigenPro 4 converges in less than 10% of the epochs needed by other methods, while also delivering superior model performance. Figure 3 compares EigenPro 4 and EigenPro 3 across multiple training epochs, following the experimental setup established in [1]. Despite EigenPro 4's linear time complexity per iteration (compared to EigenPro 3's quadratic complexity), it demonstrates faster convergence with fewer epochs. This efficiency gain is particularly pronounced for larger model sizes, where EigenPro 4 maintains or exceeds EigenPro 3's accuracy while requiring significantly fewer epochs. These results empirically validate that EigenPro 4's algorithmic improvements translate to practical benefits: not only does each iteration run faster, but fewer iterations are needed to achieve optimal performance across diverse datasets. This empirically shows that our model has a linear time complexity with respect to the model size.

## 5 Conclusion

In this work, we introduced EigenPro 4, an advancement in training large kernel models that achieves linear time complexity per iteration and linear memory scaling with model size. By implementing a delayed projection strategy, we addressed the high computational overhead previously associated with frequent projections, achieving significant time and memory efficiency improvements over EigenPro 3 and Falkon. Our empirical results on diverse datasets highlight EigenPro 4 ability to match or exceed the performance of prior methods with vastly reduced computational resources. Specifically, the algorithm demonstrates both faster convergence and superior scalability, enabling training with model sizes and datasets that were previously infeasible due to memory and time constraints.

Furthermore, EigenPro 4 design opens up new possibilities for parallelization, as it is well-suited for multi-GPU and distributed architectures. Future work will explore these aspects, further expanding its potential in real-world applications requiring efficient, scalable kernel methods for massive data volumes.

**Acknowledgements:** We acknowledge support from the National Science Foundation (NSF) and the Simons Foundation for the Collaboration on the Theoretical Foundations of Deep Learning through awards DMS-2031883 and #814639, the TILOS institute (NSF CCF-2112665), and the Office of Naval Research (N8644-NV-ONR). This work used ACCESS (Advanced cyberinfrastructure coordination ecosystem: services & support) which is supported by NSF grants numbers #2138259, #2138286, #2138307, #2137603, and #2138296. Specifically, we used the resources from SDSC Expanse GPU compute nodes, and NCSA Delta system, via allocations TG-CIS220009. This work was done in part while AA was visiting the Simons Institute for the Theory of Computing. PP was supported by the DST INSPIRE Faculty Fellowship, Schmidt Sciences, and gifts from SBI and Amazon.

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

# A Convergence analysis

In this section, we derive EigenPro4.0-Exact (Algorithm 2), a precursor to EigenPro4.0. However, this version does not scale efficiently. In Section 3, we enhance its scalability by introducing stochastic approximations, resulting in EigenPro4.0 (Algorithm 1).

Recall that the derivatives of the loss function, as defined in (13), lie in the span of the training data, denoted as $\mathcal{X}$. However, these derivatives cannot directly update the model, which resides in the span of the model centers, $\mathcal{Z}$. To address this, we first fit the labels within the $\mathcal{X}$ and then project the solution into the $\mathcal{Z}$. This process is repeated iteratively on the residual labels until convergence, as outlined in Algorithm algorithm 2.

---

**Algorithm 2** EigenPro 4-Exact

---

**Require:** Data $(X, \boldsymbol{y})$, centers $Z$
1:  $\tilde{\boldsymbol{y}}_0 = \boldsymbol{y}$
2:  **for** $t = 1, 2, \dots$ **do**
3:      $\boldsymbol{\alpha}_t = K^{-1}(X, X)\tilde{\boldsymbol{y}}_t$
4:      $K(\cdot, Z)\boldsymbol{\beta}_t = \mathrm{proj}_{\mathcal{Z}}\left(K(\cdot, X)\boldsymbol{\alpha}_t\right)$
5:      $\tilde{\boldsymbol{y}}_{t+1} = \boldsymbol{y} - K(X, Z)\boldsymbol{\beta}_t$
6:  **end for**

---

The following proposition provides the fixed point analysis for this algorithm.

**Proposition 1.** *Consider any dataset $X, \boldsymbol{y}$ and a choice of model centers $Z$, with a kernel function $K : \mathbb{R}^d \times \mathbb{R}^d \to \mathbb{R}$. Assume that $K(X, X)$ and $K(Z, X)$ are full Rank. Then, Algorithm 2 converges to the following solution:*

$$\hat{f} = K(\cdot, Z)\left(K(Z, X)K^{-1}(X, X)K(X, Z)\right)^{-1}K(Z, X)K^{-1}(X, X)\boldsymbol{y}. \tag{21}$$

*Furthermore, if $\boldsymbol{y} = K(X, Z)\boldsymbol{\beta}^* + \boldsymbol{\xi}$, where $\boldsymbol{\xi}$ is a vector of independent centered random noise with $\mathbb{E}[\xi_i^2] = \sigma^2$, then*

$$\lim_{t \to \infty} \mathbb{E}[\boldsymbol{\beta}_t] = \boldsymbol{\beta}^*, \quad \lim_{t \to \infty} \frac{\mathbb{E}[\|\boldsymbol{\beta}_t - \boldsymbol{\beta}^*\|^2]}{\sigma^2} =$$
$$\mathrm{tr}\left(\left(K(Z, X)K^{-1}(X, X)K(X, Z)\right)^{-2}K(Z, X)K^{-2}(X, X)K(X, Z)\right).$$

*Proof.* We begin by expressing Algorithm 2 recursively and substituting $\mathrm{proj}_{\mathcal{Z}}$ with the expression in (18). Recall that $f_t = K(\cdot, Z)\boldsymbol{\beta}_t$ with base case $\boldsymbol{\beta}_0 = 0$. The update rule for $\boldsymbol{\beta}_t$ is given by:

$$\boldsymbol{\beta}_t = K^{-1}(Z, Z)K(Z, X)K^{-1}(X, X)(\boldsymbol{y} - K(X, Z)\boldsymbol{\beta}_{t-1}) + \boldsymbol{\beta}_{t-1}. \tag{22}$$

Let us define the matrices:

$$B := K^{-1}(Z, Z)K(Z, X)K^{-1}(X, X), \quad C := BK(X, Z) - I,$$

which allows us to rewrite the recursion more succinctly:

$$\begin{aligned}
\boldsymbol{\beta}_t &= B(\boldsymbol{y} - K(X, Z)\boldsymbol{\beta}_{t-1}) + \boldsymbol{\beta}_{t-1} \\
&= B\boldsymbol{y} - C\boldsymbol{\beta}_{t-1} = B\boldsymbol{y} - CB\boldsymbol{y} + C^2\boldsymbol{\beta}_{t-2} \\
&\vdots \\
&= \left(\sum_{i=0}^{t-1}(-1)^i C^i\right)B\boldsymbol{y}.
\end{aligned} \tag{23}$$

As the number of iterations tends to infinity, we can define the infinite series sum:

$$S := \sum_{i=0}^{\infty}(-1)^i C^i.$$

Observe that:
$$S + CS = I.$$

Substituting the definition $C = BK(X, Z) - I$ and $B = K^{-1}(Z, Z)K(Z, X)K^{-1}(X, X)$, we have:
$$K^{-1}(Z, Z)K(Z, X)K^{-1}(X, X)K(X, Z)S = I.$$

Thus, this simplifies to:
$$S = \left(K(Z, X)K^{-1}(X, X)K(X, Z)\right)^{-1} K(Z, Z).$$

Therefore, the final solution converges to:
$$\hat{f} = K(\cdot, Z)\left(K(Z, X)K^{-1}(X, X)K(X, Z)\right)^{-1} K(Z, X)K^{-1}(X, X)\boldsymbol{y}. \tag{24}$$

Substituting $\boldsymbol{y} = K(X, Z)\boldsymbol{\beta}^* + \boldsymbol{\xi}$ readily completes the second claim.

$\square$

| line in Algorithm 1 | computation | flops |
|---|---|---|
| 7 | $K(X_m, Z)\boldsymbol{\alpha} - \boldsymbol{y}_t$ | $mp$ |
| 7 | $K(X_m, Z_{\mathsf{tmp}})\boldsymbol{\alpha}_{\mathsf{tmp}}$ | $m^2(t - kT - 1)$ |
| 7 | $K(X_m, X_s)\boldsymbol{\alpha}_s$ | $ms$ |
| 9,10 | $\boldsymbol{h}_1 := \boldsymbol{F}^\top K(X_s, X_m)\boldsymbol{g}_m \in \mathbb{R}^q$ | $ms + sq$ |
| 9 | $\boldsymbol{F}\boldsymbol{h}_1$ | $sq$ |
| 10 | $K(Z, X_m)\boldsymbol{g}_m$ | $mp$ |
| 10 | $\boldsymbol{M}\boldsymbol{h}_1$ | $pq$ |

Table 2: Computational cost analysis of Algorithm 1 for processing batch $t$ for $kT < t \le (k+1)T$ for some $k \in \mathbb{N}$.

The cost of processing batch $t$ without the post-processing adds up to $2mp + 2ms + 2sq + pq + m^2(t - kT - 1)$ flops.

## B    Computational complexity comparison

We assume that EigenPro4 is processing $T$ batches of data at once before running the post-processing step of projection. Here we show we calculated the optimal value of $T$.

**Cost for processing $t^{\text{th}}$ batch of data.**    For a some $k \in \mathbb{N}$, let $kT < t \le (k+1)T$. See Table 2.

**Cost of processing $T$ batches of data before post-processing**    The total cost for processing $T$ batches $t = kT + 1$ to $t = (k+1)T$ before the projection is the sum of the above

$$T(2mp + 2ms + 2sq + pq) + m^2 \sum_{t=kT+1}^{(k+1)T} (t - kT - 1) = T(2mp + 2ms + 2sq + pq) + m^2 \frac{T(T-1)}{2} \tag{25}$$

**Average of processing $T$ batches of data with post-processing**    Assuming the post processing involves $T_{\mathsf{ep2}}$ epochs of EigenPro 2, the average cost of processing $T$ batches is

$$\frac{T(2mp + 2ms + 2sq + pq) + m^2 \frac{T(T-1)}{2} + p^2 T_{\mathsf{ep2}}}{T} \tag{26}$$

A simple calculation shows that

$$T^\star = \frac{p}{m}\sqrt{2T_{\mathsf{ep2}}} \tag{27}$$

minimizes the average time above. The average cost of processing a batch is thus

$$2mp(1 + \sqrt{2T_{\mathsf{ep2}}}) + 2ms + 2sq + pq \tag{28}$$

**Illustration for delayed projection for $T = 4$.**

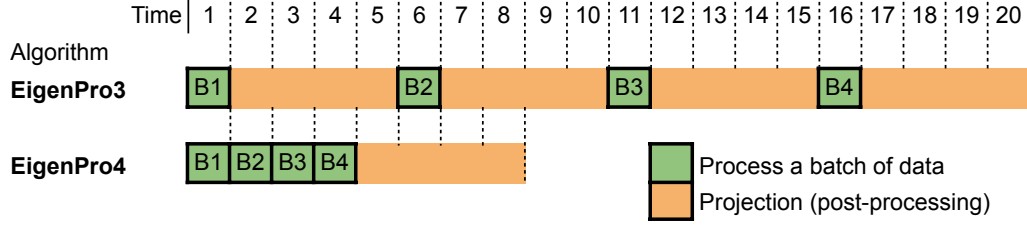

Figure 4: **Design of EigenPro4.** An illustration of how batches of data are processed by the two algorithms. EigenPro3 involves an expensive *projection* step when processing every batch of data. EigenPro4 waits for multiple batches to be processed before running the projection step for all of them together. This reduces the amortized cost for processing each batch.

**Comparison between EigenPro 4 and EigenPro 3.** Figure 5 shows how EigenPro 4 and EigenPro 3 perform over training iterations. EigenPro 4 accuracy improves between projections and drops after

each projection step. While EigenPro 3 projects at every step, EigenPro 4 maintains comparable accuracy with fewer projections. The left panel of Figure 5 confirms that both methods reach similar final accuracy, while the right panel shows EigenPro 4 significant speed advantage. With continued training, EigenPro 4 accuracy drops from projections become progressively smaller.

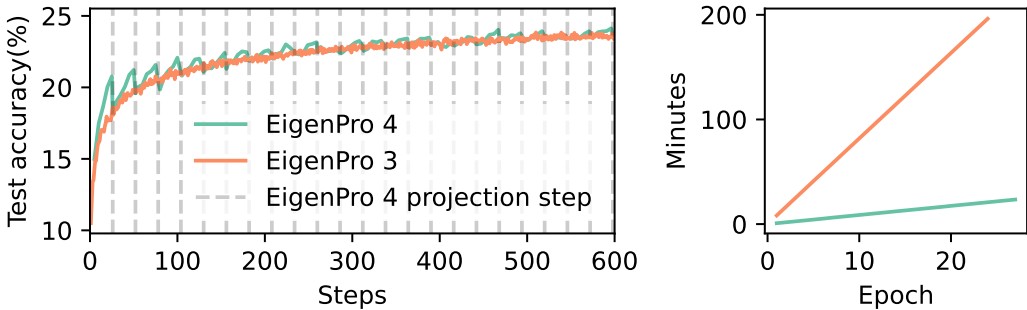

Figure 5: Performance and computational time comparison between EigenPro 4.0 ($T = 11$) and EigenPro 3.0 (equivalent to $T = 1$), highlighting the impact of the projection step on the performance of EigenPro 4.0. The detail of the experiment can be found in Appendix C.

| Algorithm | setup | FLOPS per batch* | Memory |
|---|---|---|---|
| EigenPro 4.0 | $O(s^2q)$ | $2mp(1 + \sqrt{2T_{\text{ep2}}}) + 2ms + 2sq + pq$ | $s^2 + p(1 + \sqrt{2T_{\text{ep2}}})$ |
| EigenPro 3.0 | $O(s^2q)$ | $2mp + p^2 T_{\text{ep2}} + 2ms + 2sq + pq$ | $s^2 + p$ |
| Falkon | $O(p^3)$ | $2mp$ | $p^2$ |

Table 3: **Comparing complexity of algorithms.** Number of training samples $n$, number of model centers $p$, batch size $m$, Nyström sub-sample size $s$, preconditioner level $q$. Here we assumed only a constant number of epochs of EigenPro 2.0 is needed for large scale experiments. Cost of kernel evaluations and number of classes are assumed to be $O(1)$, also it is reasonable to assume $p \gg s \gg q$. * FLOPS per iteration reported are amortized over multiple batches processed.

# C Experiments Results

## C.1 Other kernels

For consistency with prior work, we primarily used the Laplace kernel, which offers efficient per-batch computation and scales linearly with total training time—making it feasible to train methods such as EigenPro 3.0 at larger scales. For completeness, we also report results using alternative kernels: Gaussian (bandwidth 20) and NTK (ReLU MLP, depth 2), evaluated on the same CIFAR5M embeddings as in Figure 1.

Table 4: Comparison of kernel types and number of centers.

| Kernel Type | 64k Centers | 128k Centers | 256k Centers |
|---|---|---|---|
| Gaussian | 87.77% (178s) | 87.80% (194s) | 88.14% (233s) |
| NTK | 87.01% (206s) | 87.34% (309s) | 87.54% (564s) |

## C.2 Computational resources used

This work used the Extreme Science and Engineering Discovery Environment (XSEDE) [23]. We used machines with NVIDIA-V100, NVIDIA-A100 and NVIDIA-A40 GPUs, with a V-RAM up to 1.3 T, and 8x cores of Intel(R) Xeon(R) Gold 6248 CPU @ 2.50GHz with a RAM of 100 GB. NOte that we had 1.3T of RAM for just one experiment CIFAR5M*, for the rest of expermients we where constraint with 400G of RAM.

## C.3 Datasets

We perform experiments on these datasets: (1) CIFAR10, [11], (2) CIFAR5M, [17], (3) ImageNet, (4) Webvision.[12], and (5) librispeech.

**CIFAR5M.** In our experiments, we utilized both raw and embedded features from the CIFAR5M data-set. The embedded features were extracted using a MobileNetv2 model pre-trained on the ImageNet data-set, obtained from *timm* library [25]. We indicate in our results when pre-trained features were used by adding an asterisk (*) to the corresponding entries.

**ImageNet.** In our experiments, we utilized embedded features from the ImageNet data-set. The embedded features were extracted using a MobileNetv2 model pre-trained on the ImageNet dataset, obtained from *timm* library [25]. We indicate in our results when pre-trained features were used by adding an asterisk (*) to the corresponding entries.

**Webvision.** In our experiments, we utilized embedded features from the Webvision data-set. The embedded features were extracted using a ResNet-18 model pre-trained on the ImageNet dataset, obtained from *timm* library [25]. Webvision data set contains 16M images in 5K classes. However, we used only a subset of it for different experiments.

**Important note about Webvision dataset.** In the experiments shown in Figure 3, to ensure computational feasibility for running EigenPro 3.0, we used only data with labels smaller than 500, corresponding to roughly 2.5 million data points. In contrast, the results in Table 1 were obtained using labels up to 1000, encompassing about 5.2 million data points. Since this larger label range represents a more challenging classification problem, the test accuracy in Figure 3 is higher than in Table 1.

**Librispeech.** Librispeech [18] is a large-scale (1000 hours in total) corpus of 16 kHz English speech derived from audio books. We choose the subset train-clean-100 and train-clean-300 (5M samples) as our training data, test-clean as our test set. The features are got by passing through a well-trained acoustic model (a VGG+BLSTM architecture in [8] ) to align the length of audio and text. It is doing a 301-wise classification task where different class represents different uni-gram [10]. The implementation of extracting features is based on the ESPnet toolkit [24].

## C.4 Experiments details

**Figure 1** This experiment used CIFAR5M* data set, where embedding has been generated using a pre-trained mobile-net network mentioned earlier. this is the only experiment that we had access to 1.3T of VRAM. We set the bandwidth to 5.0 and use $1k$ Nystrom samples with preconditioning level of size 100. We used float16 for this experiment.

**Figure 5**  This experiment has been run over Webvision data set with extracted embedding through Resnet18. the model size here is set to $100k$ number of centers. The bandwidth used is $5.0$, $1k$ Nystrom samples with preconditioning level of size $100$. We used float16 for this experiment.

**Figure 3**  We follow the setting in [1]. The bandwith used here is 20 for Librispeach and Webvision and 16 for imagnet. Here again we used extracted feature of these datasets mentioned earlier. The precision used here is float32. with $10k$ Nystrom samples with preconditioning level of size 1000.

**Table 1**  For all datasets here we used bandwidth of 5.0 with $1k$ Nystrom samples with preconditioning level of size 100. We used float16 for all dataset except for Librispeach where we used float32. Further, we note that Falkonlatest library ran out of GPU memory for model sizes larger than 256000 number of centers that is the reason we could not run it for 256000. And as mentioned for model sizes 521000 and above the algorithm has inherent quadratic scaling with respect to model size and we ran out of VRAM. In the plot we refer to both of these memry issues as OOM.

