# OpenReview forum: "Fast Training of Large Kernel Models with Delayed Projections"
_NeurIPS.cc/2025/Conference — NeurIPS 2025 spotlight_

### Official Review · Reviewer_tAZu · 2025-06-03

**Clarity:** 4
**Significance:** 3
**Originality:** 2
**Rating:** 5
**Confidence:** 5

**Summary:**

The authors propose EigenPro4, the algorithm for fast training of kernel machines using Preconditioned Stochastic Gradient Descent. This work builds upon the previous versions of the algorithms: EigenPro, EigenPro2, and EigenPro3. The key trick is to use a subset of training points to represent the function, rather than using all trainingp points. The key difficulty is that the computed gradient in each iteration lies in the span of the training points, not in the span of the chosen Nystrom centers. In order to maintain the iterate remains in the span of the Nystrom centers (low rank), a projection method which is the most expensive step ($O(p^2)$ per iterate) was proposed in EigenPro3. This projection step was improved to $O(p)$ in EigenPro4 by delaying the projections for $T$ iterations. This means in-between the two successive projection steps, the prediction function has additional temporary centers; so for $T$ delayed projections, the maintained centers would be expanded by at most $T m$ additional centers. The authors compare EigenPro4 against EigenPro3 and Falkon on real-world datasets.

**Questions:**

1. I could not find what parameters were used for EigenPro4 in the experiment. Could the authors clarify?
2. In general, the results for each method used should be parameterized (e.g. Table 1). In general, it is hard to find these parameters within the writeup.
3. Would it be possible to incorporate a nearest neighbor heuristic within Algorithm 1 (Line 8) to further maintain a more compress set of centers?
4. Other than the Laplace kernel (again, I was not able to find which kernel was being used for the experiment in the captions), what other kernels have the authors tried?

**Ethical Concerns:**

["NO or VERY MINOR ethics concerns only"]

**Final Justification:**

In light of the authors' responses and technical presentation of the paper, I am recommending the paper for acceptance. While the extensions could be seen as obvious, I think this paper does deserve a spot as it has applications in which kernel methods (and may also provide some inspiration of scaling deep neural networks) are applicable.

**Limitations:**

There are no significant limitations in the authors' work. However, while the experimental results for the most part are clearly presented, the authors are encouraged to tabulate the parameters used for each method for clarity.

**Paper Formatting Concerns:**

There are no major formatting concerns.

**Quality:**

3

**Strengths And Weaknesses:**

Strengths: The manuscript is clearly written and the authors have actually made conscientious efforts to limit unnecessarily complex notations to aid in understanding the algorithm. Experimental evaluations are thorough and enlightening, although they do not cover every variant of fast kernel machines.

Weaknesses: The improvement is rather trivial and relies on delaying the projection step to improve efficiency of the algorithm. But perhaps this triviality makes the manuscript stand out in times when many obscure papers are abundant.

---

> ### Author Rebuttal · Authors · 2025-07-30
>
> We thank the reviewer for their positive comments. We address the concerns one by one as follows:
>
> >I could not find what parameters were used for EigenPro 4 in the experiment. Could the authors clarify?
>
> The hyper-parameters used in our experiments are listed in Appendix Section C. If there is a specific hyper-parameter you are referring to, we would be happy to clarify further.
>
> >In general, the results for each method used should be parameterized (e.g., Table 1). In general, it is hard to find these parameters within the writeup.
>
> Thank you for the suggestion. If you are referring to the hyper-parameters for each method, we will add a comprehensive table summarizing all hyper-parameters across experiments and methods in the final version to improve readability.
>
> >Would it be possible to incorporate a nearest neighbor heuristic within Algorithm 1 (Line 8) to further maintain a more compressed set of centers?
>
> How to add or choose centers is indeed an important future direction. One possible approach is to use clustering techniques like K-means to avoid selecting redundant centers. Incorporating nearest neighbor heuristics, as you suggested, is another promising direction to maintain a more compact and diverse set of centers.
>
> >Other than the Laplace kernel (again, I was not able to find which kernel was being used for the experiment in the captions), what other kernels have the authors tried?
>
> We used the Laplace kernel in our experiments, primarily for consistency with prior work and because its per-batch computation is highly efficient. Specifically, the total training time scales linearly with the per-batch kernel computation time, which makes it feasible to explore more computationally intensive methods like EP3.0—especially at larger model scales. For completeness, we will include results using the NTK kernel for EP4.0 in the final version.
>
> Some results using alternative kernels: Gaussian (with bandwidth 20) and NTK (ReLU MLP with depth 2), evaluated on the same extracted embeddings from the CIFAR5M dataset as used in the main paper.
> | Kernel Type | 64k Centers       | 128k Centers      | 256k Centers      |
> |-------------|-------------------|-------------------|-------------------|
> | Gaussian    | 87.77% (178s)     | 87.80% (194s)     | 88.14% (233s)     |
> | NTK         | 87.01% (206s)     | 87.34% (309s)     | 87.54% (564s)     |

---

> > ### Comment · Reviewer_tAZu · 2025-08-02
> > **Dear authors**
> >
> > Thank you for your responses; I am satisfied with the responses and currently will be advocating for acceptance. I am looking forward to engaging in further discussion with other reviewers.

---

> > > ### Author Response · Authors · 2025-08-04
> > > **Thank you for championing our paper**
> > >
> > > -

---

### Official Review · Reviewer_aQLu · 2025-06-10

**Clarity:** 2
**Significance:** 3
**Originality:** 2
**Rating:** 4
**Confidence:** 3

**Summary:**

The paper proposes a novel algorithm EigenPro 4 for training large kernel models.
The algorithm is an upgrade an existing EigenPro 3 version which uses preconditioned stochastic gradient descent with certain projections in order to decouple the model size from the training set size. This projection requires quadratic time in the model size and so far had to be computed after each stochastic step which yields the bottleneck for the running time.
The authors overcome this bottleneck by delaying the projections for $T$ iterations, where $T$ is a predefined hyperparameter (it is claimed that the optimal value for $T$ is the ratio of model size and batch size).

They validate their approach empirically with experiments on three datasets which show a clear speedup of the EigenPro 4 version compared to EigenPro3 and another state-of-the art method.

**Questions:**

1) You mention that you prove convergence of Alg 1 for $T=\infty$ in the appendix. This would mean that it never projects, which does not really make sense. In fact, you prove it for $T=1$?

2) How did you choose $T$ and all other hyperparameters in your experiments?

3) Do you have an explanation why also the convergence rate is faster?

4) Can your algorithm compete with other models?

**Ethical Concerns:**

["NO or VERY MINOR ethics concerns only"]

**Final Justification:**

After considering the rebuttal and also the comments by the other reviewers, I find the paper to be acceptable.

**Limitations:**

The paper does not really discuss any limitations of the approach.
For example, how does the algorithm perform in comparison to other (non-kernel) machine learning methods.

**Quality:**

3

**Strengths And Weaknesses:**

Strength: The paper presents a natural approach to speed up the computational performance of the state-of-the art of kernel based learning. The performance seems to be considerably improved at least on the investigated datasets. Also the convergence rate was faster.

Weakness: The result is very incremental. The idea to delay projections seems rather obvious and of little originality. The increase in performance looks solid, but I am not sure about the significance for practical purposes (since I am not an expert in practical applications).
Also, there is no theoretical analysis provided that guarantees convergence.
Finally, the presentation could be improved. Many things are not defined or explained and there are several language errors. (See comments below.)

Comments:
- l 28: "to improve condition number" -> improve the condition number
- l 51: "on CIFARM data set" -> on the CIFARM data set
- l 63: z_i not mathbold. Also explain the idea behind these centers of a general kernel model.
- l 65: Define what a RKHS is.
- Fig 1: Explain notation p.
- l 80:  Explain the tensor product sign.
- l 95: "condition number is ill conditioned" -> This is a confusing formulation.
- l 102: "unit-norm e_i" -> Do you mean standard unit vector?
- l 104: "let ... as the" -> let ... be the
- l 134: "models of this form" -> which form are you referring to?
- l 136: The variable m is not defined yet. (Also you use m later for mini batches (X_m,y_m), which could be confusing)
- l 150: "add to Z_tmp with temporary centers" -> add to Z_tmp the temporary...
- Fig 3: What is n?

---

> ### Author Rebuttal · Authors · 2025-07-30
>
> We thank the reviewer for their feedback. We address the concerns one by one as follows:
>
>
>
> > The increase in performance looks solid, but I am not sure about the significance for practical purposes (since I am not an expert in practical applications).
>
> Kernel methods play a crucial role in understanding modern learning phenomena such as feature learning—an area that has gained renewed attention, including in recent work published in *Science* [1]. We will revise the paper to include a clearer and more detailed motivation for studying kernel models and their relevance.
>
> [1] Radhakrishnan, A., Beaglehole, D., Pandit, P., & Belkin, M.
> *Mechanism for feature learning in neural networks and backpropagation-free machine learning models.*
> Science, 383(6690), 1461–1467, 2024.
>
>
>
> > Also, there is no theoretical analysis provided that guarantees convergence.
>
> We provide a theoretical analysis in the appendix for the special case when $T = \infty$. However, the primary focus of this paper is to demonstrate the empirical effectiveness of our method across a range of datasets, in comparison to prior state-of-the-art approaches.
>
>
>
> > You mention that you prove convergence of Alg 1 for $T = \infty$ in the appendix. This would mean that it never projects, which does not really make sense. In fact, you prove it for?
>
> To clarify, $T = \infty$ does not mean that we never project. Rather, it refers to an idealized setting in which temporary centers are added infinitely many times. Since the training set is finite and temporary centers are selected from this set, the process can be represented by assigning weights to each data point. We know these weights converge to the closed-form solution:
>
> $$
> \alpha^* = K(X, X)^{-1} y
> $$
>
> Once this limiting solution is reached for $T = \infty$, we project the resulting model back to the original model with a fixed set of centers. This process is then repeated iteratively. Our analysis focuses on this limiting behavior to better understand the convergence dynamics of the algorithm.
>
>
>
> > How did you choose $T$ and all other hyperparameters in your experiments?
>
> For most hyperparameters, such as the kernel bandwidth, we followed standard choices from prior work—particularly those used in EigenPro 3. Similarly, for the preconditioner, we adopted the typical setting where the ratio of the number of Nyström samples to the number of top eigenvalues to suppress is around 10–15. The only new hyperparameter introduced in our method is the projection frequency $T$, which we discuss in Appendix Section B. There, we explain that $T$ should scale with the ratio of the model size $p$ to the batch size $m$.
>
>
>
> > Do you have an explanation why also the convergence rate is faster?
>
> If you are referring to Figure 3, that was an empirical observation.
>
>
>
> > Can your algorithm compete with other models?
>
> Our paper specifically focuses on the computational complexity of kernel models, rather than benchmarking performance against non-kernel models. To this end, we compare our method to two prior state-of-the-art solvers for kernel methods, which is the relevant baseline for our study.
>
>
>
> > Comments:
>
> Thank you for pointing these out. We will fix them in the final version of the paper.

---

> > ### Comment · Reviewer_aQLu · 2025-08-04
> >
> > Thank you for your response. Based on your clarifications and the other reviews, I increase my score.

---

### Official Review · Reviewer_CFJP · 2025-06-25

**Clarity:** 2
**Significance:** 2
**Originality:** 2
**Rating:** 4
**Confidence:** 3

**Summary:**

This work (EigenPro 4) proposes Delayed Projection, instead of projection after each gradient descent step, to address high time cost per batch of EigenPro 3. Convergence and time complexity analysis are given. Experiments show the great reduction in per-epoch training time and good scalability w.r.t. model size.

**Questions:**

See Weakness.

**Ethical Concerns:**

["NO or VERY MINOR ethics concerns only"]

**Final Justification:**

The recommended score of 4 is maintained. While the work demonstrates strong technical merit and the authors have effectively addressed all the concerns during the rebuttal, a slight lack of novelty leads to this nuanced recommendation.

The motivation for delayed projection is clear and intuitive, directly addressing the high per-batch time cost of EigenPro 3. The experiments results are convincing and demonstrate the reduced per-epoch training time and good scalability with respect to model size. And the time reduction does not cause severe test accuracy drop.

During the rebuttal, the authors addressed all my concerns. They clarified the relation between model size and EigenPro 4’s performance, and explained the reason of accuracy change in the convergence process. My misinterpretation of accuracy drops is addressed with clear explanation.

The only mitigating factor is a slight lack of novelty in the core "delayed projection" mechanism. The concept of delaying projections, as a strategy to balance efficiency and performance, is built upon existing projection-based optimization paradigms rather than introducing a paradigm-shifting innovation. This is not a flaw, but a subtle limitation that prevents a stronger recommendation.

I would assign most of my weight on the work’s clear motivation, sound theoretical analysis and convincing experiments. I would assign little weight on the slight lack of novelty. The borderline accept recommendation comes not from weaknesses, but from a subtle lack of novelty. However, this should not reduce the significance of the authors’ contribution. The proposed method meaningfully advances EigenPro’s efficiency.

**Limitations:**

yes

**Quality:**

3

**Strengths And Weaknesses:**

Strengths:

1.	The motivation of delayed projection is clear and easy to understand.

2.	The analysis of convergence and time complexity is thorough and sound.

3.	The experiment results demonstrate that the proposed method is practical on very large datasets . Compared with EigenPro 3, the time reduction does not cause severe test accuracy drop.

Weaknesses:

1.	In terms of effectiveness, EigenPro 4 performs inconsistently across different model sizes: it is strange to observe that although EigenPro 4 is more efficient than EigenPro 3, the accuracy of EigendPro 4 is lower (higher) than EigenPro 3 when the model size is small (large). Is there any particular reason behind this?

2.	Although EigenPro 4 is faster than EigenPro 3 and converges faster. The results in Figure 5 indicate the convergence process of EigenPro 4 is more unstable than EigenPro 3 due to delayed projection, especially in Webvision dataset. Is this because EigenPro 4 accuracy improves between projections and drops after each projection step as shown in Figure 5?

3.	As claimed in line 727, the EigenPro 4 accuracy drops from projections become progressively smaller. However, in the Webvision and Librispeech results in Figure 5, the multiple accuracy drops remain relatively large when EigenPro 4 converges. Is there any specific reason behind this occurrence that warrants further exploration on the stability of EigenPro 4? Adding an analysis on this issue would be nice.

---

> ### Author Rebuttal · Authors · 2025-07-30
>
> We thank the reviewer for their feedback. We address the concerns one by one as follows:
>
> >In terms of effectiveness, EigenPro 4 performs inconsistently across different model sizes: it is strange to observe that although EigenPro 4 is more efficient than EigenPro 3, the accuracy of EigenPro 4 is lower (higher) than EigenPro 3 when the model size is small (large). Is there any particular reason behind this?
>
> From a generalization perspective, EigenPro 4 and EigenPro 3 converge to different solutions, and their exact generalization behavior depends on the specific problem. However, we note that EigenPro 4.0 tends to be more effective at larger model sizes. This is because it allows for longer delays between projections(for a fixed batch size), and each projection step discards less information—the larger model has a better chance of adequately covering the relevant function space.
>
> >Although EigenPro 4 is faster than EigenPro 3 and converges faster. The results in Figure 5 indicate the convergence process of EigenPro 4 is more unstable than EigenPro 3 due to delayed projection, especially in Webvision dataset. Is this because EigenPro 4 accuracy improves between projections and drops after each projection step as shown in Figure 5?
>
> In the EP4 algorithm, the model expands between projections by incorporating temporary centers—it is not a fixed model during this phase. After each projection step, the model reverts to its original form with a fixed set of centers. This explains the accuracy drop observed after each projection step in Figure 5: the temporary expansion improves accuracy, but once the projection is applied, the model returns to its constrained representation.
>
> >As claimed in line 727, the EigenPro 4 accuracy drops from projections become progressively smaller. However, in the Webvision and Librispeech results in Figure 5, the multiple accuracy drops remain relatively large when EigenPro 4 converges. Is there any specific reason behind this occurrence that warrants further exploration on the stability of EigenPro 4? Adding an analysis on this issue would be nice.
>
> We believe you may be referring to Figure 3, rather than Figure 5. To clarify, Figure 5 plots accuracy over training steps, where projection steps are explicitly marked with dashed vertical lines. In contrast, Figure 3 shows accuracy over training epochs, where a projection occurs at the end of each epoch (In addition to every T steps projection).

---

> > ### Comment · Reviewer_CFJP · 2025-08-04
> >
> > Thank you for your responses. I am satisfied that the responses address my concrens. I keep my positive grading as is for now and will engage in discussion with other reviewers. I currently advocate for acceptance.

---

### Official Review · Reviewer_TF8r · 2025-06-25

**Clarity:** 2
**Significance:** 3
**Originality:** 2
**Rating:** 4
**Confidence:** 4

**Summary:**

Previous kernel-based methods have difficulties with scaling to larger model sizes due to the computational cost of projecting gradients into the kernel feature space after every iteration. The authors propose EigenPro 4, which improves upon EigenPro 3, by delaying this projection by T iterations, which reduces the per iteration cost while maintaining performance. This approach also allows the model to be approximately linear on average per epoch in terms of per iteration computations while having linear memory scaling.

**Questions:**

The motivation needs improvement a lot, such as why to choose a kernel-based method over a non-kernel based method (e.g. deep learning models)?

In equation 15, the otherwise branch calculates the stochastic gradient using $f_t$, but should it not be on $f_{t-1}$?

**Ethical Concerns:**

["NO or VERY MINOR ethics concerns only"]

**Final Justification:**

The rebuttal addressed my concerns, and I'd love to raise my score.

**Limitations:**

Yes

**Quality:**

3

**Strengths And Weaknesses:**

The paper presents a practical contribution to kernel methods which would help allow them to be used in more large scaled environments due to the linear amortized per-iteration cost. They also show impressive performance improvements over EigenPro 3 and Falkon in terms of running time. However, the paper itself, especially the beginning sections, are not the most organized nor standalone. There is little contextualization in the beginning, and no clear reason as to why one would want to use kernel methods even with linear amortized per iteration cost and linear memory scaling. Although, the contributions could be significant, the paper does a poor job phrasing its importance with respect to other non-kernel methods. There are also some things missing that would make for a comphrensive evaluation, such as using more kernels in the addition to the laplace kernel or abalations on the value of $T$. Although, $T$ is proportional to $p/m$, no exact value used is provided for the experiments and testing different values for $T$ would be informative.

---

> ### Author Rebuttal · Authors · 2025-07-30
>
> We thank the reviewer for their feedback. We address the concerns one by one as follows:
>
> >Although the contributions could be significant, the paper does a poor job phrasing its importance with respect to other non-kernel methods.
>
> Our paper is specifically focused on the computational complexity of kernel models, rather than their performance in comparison to non-kernel methods. Kernel methods play a crucial role in understanding modern learning phenomena such as feature learning – an area that has gained renewed attention, including highly performant variants of kernel machines which recently appeared in Science [1]. We will revise the paper to include a clearer and more detailed motivation for studying kernel models and their relevance.
> [1] Radhakrishnan, A., Beaglehole, D., Pandit, P., & Belkin, M.
>  Mechanism for feature learning in neural networks and backpropagation-free machine learning models.
>  Science, 383(6690), 1461–1467, 2024
>
>
>  >There are also some things missing that would make for a comprehensive evaluation, such as using more kernels in the addition to the laplace kernel or ablations on the value of .
>
> We chose the Laplace kernel primarily for consistency with prior work and because its per-batch computation is highly efficient. Specifically, the total training time scales linearly with the per-batch kernel computation time, which makes it feasible to explore more computationally intensive methods like EP3.0—particularly for larger model sizes. For completeness, we will include results with the NTK kernel in the final version for EP4.0.
>
> Some results using alternative kernels: Gaussian (with bandwidth 20) and NTK (ReLU MLP with depth 2), evaluated on the same extracted embeddings from the CIFAR5M dataset as used in the main paper.
> | Kernel Type | 64k Centers       | 128k Centers      | 256k Centers      |
> |-------------|-------------------|-------------------|-------------------|
> | Gaussian    | 87.77% (178s)     | 87.80% (194s)     | 88.14% (233s)     |
> | NTK         | 87.01% (206s)     | 87.34% (309s)     | 87.54% (564s)     |
>
> >In equation 15, the otherwise branch calculates the stochastic gradient using , but should it not be on ?
>
> Could you please clarify your question regarding Equation 15? It seems the sentence was cut off, and the specific expressions you're referring to are missing. We're happy to address the concern once we have the full context.

---

> > ### Comment · Reviewer_TF8r · 2025-08-04
> >
> > Thanks for the authors' rebuttal. You should be able to read my question in my review, as I can see it. Anyway, my question is: In equation 15, the otherwise branch calculates the stochastic gradient using $f_t$, but should it not be on $f_{t-1}$?

---

> > > ### Author Response · Authors · 2025-08-04
> > >
> > > We apologize for missing the mention of \( f_{t-1} \) and \( f_t \) in your comment. That is a typo—it should be \( f_{t-1} \). We will correct this in the final version. Thank you for pointing it out.

---

> > > > ### Comment · Reviewer_TF8r · 2025-08-04
> > > >
> > > > After reading the rebuttal and the comments from the other reviewers, I'd like to raise my score.

---

### Official Review · Reviewer_qGfb · 2025-06-26

**Clarity:** 2
**Significance:** 4
**Originality:** 2
**Rating:** 5
**Confidence:** 2

**Summary:**

The authors propose a new method for solving kernel regression via gradient
descent. Their method, EigenPro 4, is based heavily upon EigenPro 3. The paper
identifies the 'projection step' as a major bottleneck in EigenPro 3, and solve
this issue by amortization: instead of projecting every step, they accumulate 'temporary centres'
and project them every T steps. This modification leads to massive speedups vs.
EigenPro 3, and (perhaps surprisingly) also improves performance. The speedups
are especially significant, as this unlocks the ability to use kernel regression
for larger datasets than previously practicable.

**Questions:**

- Why do we get such a noticable accuracy/performance boost for EP4 over EP3 in
  Figure 3, when the only difference between the two is the amortization of the
  projection? Can you comment on whether the amortization might improve
  properties of the optimization?
- Figure 5 shows that performance degrades after the projection step. Are you
  able to explain this behaviour?
- Can you provide any preliminary experiments with other kernels (i.e. not just
  the Laplace kernel)? You mentioned that Laplace has strong empirical
  performance, but compared to what baseline?
- Is it possible to perform 'deep kernel' experiments? I.e. to stack several
  kernel regressors together, and train them jointly with EP4?

**Ethical Concerns:**

["NO or VERY MINOR ethics concerns only"]

**Final Justification:**

In my original review, I was concerned about issues such as performance degrading when centres are consolidated, lack of results for different kernels, lack of comparison to neural networks. These have been addressed, except the comparison to NNs, but upon seeing the authors' justification for this, I think it is reasonable for it to be excluded.

I am happy to raise my score, as I think the paper is a solid contribution to kernel methods, and my issues have been resolved.

**Limitations:**

The authors discuss theoretical limitations of their algorithm in Section 3.6. However, I would like to see a discussion of practical elements too, such as where EP4 is limited versus other methods (e.g. NNs). For example, as far as I know, EP4 is relying on good underlying representations as input to be useful in practice.

**Quality:**

3

**Strengths And Weaknesses:**

Strengths:

- Authors give a clear and concise exposition of kernel regression and of existing algorithms.
- Experimental results very clearly show that EP4 is significantly faster and
  more practical than kernel regressor alternatives.

Overall EP4 is obviously very helpful for any practitioners using kernel regression / EP3.

Weaknesses:

- Experiments/Datasets are a bit artificial. For example, the kernel method uses
  features extracted from a ResNet/MobileNetV2.
- It's unclear how the proposed methods performs vs. a NN. Even if the NN is
  more performant, this still seems relevant to include, considering that the
  underlying features of all experiments are extracted from a NN.
- Some experimental details are missing in the main text; for example, basic information like what value
  of T is used?
- The paper's results suggest that EP4 is strictly better (in terms of accuracy/loss) than EP3, but more
  results using further kernels/datasets (only 1 kernel and 4 dataset were used)
  would be helpful to confirm this.
- The paper does not explain why EigenPro 4 is better than EigenPro 3, which is
  an interesting quirk. I would like to see a discussion on why accumulating the temporary centres appears to help.


Minor issues:

- Algorithm 1 describes computational details of EigenPro 4. It would be nice to
  have a 'diff' with EigenPro 3 to clarify the computational differences.
- In Eq. (1) L is defined differently to L in line 78. Make these consistent.
- Line 81, 'Operator K' -> 'The Operator K'
- The meaning of * in line 193 (and elsewhere). Please include this,
  possibly as a footnote, in the main text.

---

> ### Author Rebuttal · Authors · 2025-07-30
>
> We thank the reviewer for their feedback. We address the concerns one by one as follows:
>
> >Experiments/Datasets are a bit artificial. For example, the kernel method uses features extracted from a ResNet/MobileNetV2.
> And
> > It's unclear how the proposed method performs vs. a NN. Even if the NN is more performant, this still seems relevant to include, considering that the underlying features of all experiments are extracted from a NN.
>
> The primary focus of our paper is on the computational complexity of kernel methods, rather than on comparing their predictive performance to that of neural networks. Kernel methods continue to play an important role in understanding modern learning phenomena such as feature learning, a topic that has received renewed interest. Furthermore there are recently developed high-performing variants of kernel methods such as Recursive Feature Machines (RFM) recently published in Science [1]. We will revise the paper to include a more detailed motivation for the relevance and significance of studying kernel models in this context.
> We followed prior works in using datasets with features extracted from pretrained networks (e.g., ResNet, MobileNetV2 or VGG+BLSTM for librispeech) to enable fair and consistent comparisons with prior works. In addition, we also include results on raw image inputs from CIFAR-5M, which demonstrates that our method is not limited to feature-extracted datasets.
>
> [1] Radhakrishnan, A., Beaglehole, D., Pandit, P., & Belkin, M.
>  Mechanism for feature learning in neural networks and backpropagation-free machine learning models.
>  Science, 383(6690), 1461–1467, 2024.
>
> >Some experimental details are missing in the main text; for example, basic information like what value of $T$ is used?
>
> We provide a detailed discussion on how to choose $T$ in Appendix Section B, where we explain that it should scale as \$ \mathcal{O}(\frac{p}{m}) $, with $p$ denoting the model size and $m$ the batch size. In all experiments, we set $ T = 2 \times \frac{p}{m} $. We will clarify this choice further  in Appendix Section C in the final version.
>
> >The paper's results suggest that EP4 is strictly better (in terms of accuracy/loss) than EP3, but more results using further kernels/datasets (only 1 kernel and 4 dataset were used) would be helpful to confirm this.
>
> …
> >The paper does not explain why EigenPro 4 is better than EigenPro 3, which is an interesting quirk. I would like to see a discussion on why accumulating the temporary centres appears to help.
>
> …
> >Why do we get such a noticeable accuracy/performance boost for EP4 over EP3 in Figure 3, when the only difference between the two is the amortization of the projection? Can you comment on whether the amortization might improve properties of the optimization?
>
> We would like to clarify that EigenPro 4 and EigenPro 3 converge to potentially different solutions. From a generalization standpoint (accuracy/loss), it is not possible to claim that one is strictly better than the other – it depends on the specifics of the dataset and the problem setting. Our primary goal in this paper is to demonstrate that EigenPro 4 offers substantial speed improvements without sacrificing generalization. The observed performance boost is not the main focus, but it does indicate that the speedup does not come at the cost of degraded accuracy.
>
> As for the role of amortization and temporary centers: accumulating temporary centers between projections allows the model to explore a richer hypothesis space during intermediate steps. While the model eventually projects back to the original set of centers, these intermediate updates may help guide optimization toward better regions of the parameter space. A deeper theoretical understanding of this effect is an interesting direction for future work, and beyond the scope of the current paper.
>
> About the kernel:
> We chose the Laplace kernel primarily for consistency with prior work and because its per-batch computation is highly efficient. Specifically, the total training time scales linearly with the per-batch kernel computation time, which makes it feasible to explore more computationally intensive methods like EP3.0—particularly for larger model sizes. For completeness, we will include results using the NTK kernel in the final version of the paper.
>
> Results using alternative kernels: Gaussian (with bandwidth 20) and NTK (ReLU MLP with depth 2), evaluated on the same extracted embeddings from the CIFAR5M dataset as used in the main paper.
> | Kernel Type | 64k Centers       | 128k Centers      | 256k Centers      |
> |-------------|-------------------|-------------------|-------------------|
> | Gaussian    | 87.77% (178s)     | 87.80% (194s)     | 88.14% (233s)     |
> | NTK         | 87.01% (206s)     | 87.34% (309s)     | 87.54% (564s)     |
>
>
>
> >Figure 5 shows that performance degrades after the projection step. Are you able to explain this behaviour?
>
> In the EP4 algorithm, the model expands between projections by incorporating temporary centers—it is not a fixed model during this phase. After each projection step, the model reverts to its original form with a fixed set of centers. This explains the accuracy drop observed after each projection step in Figure 5: the temporary expansion improves accuracy, but once the projection is applied, the model returns to its constrained representation.

---

> > ### Comment · Reviewer_qGfb · 2025-08-05
> >
> > Thank you for the informative response to my review!
> >
> > I agree that NN baselines are not necessary for this work.
> >
> > While I'm convinced by most of the rebuttal, I'd like to see experiments with additional kernels beyond just NTK.
> >
> > > For completeness, we will include results using the NTK kernel in the final version of the paper.
> >
> > Since practitioners commonly use kernels other than Laplace, these experiments would better demonstrate the generalizability of your approach and provide more practical value to the community (even if just with example training scripts).

---

> > > ### Author Response · Authors · 2025-08-07
> > >
> > > We thank the reviewer and are glad we could address your concerns. Regarding kernel types, we have provided results for the most commonly used kernels: (1) Laplace, (2) Gaussian (RBF), and (3) NTK. If the reviewer has a specific suggestion, we would be happy to add it to the final version.

---

> > > > ### Comment · Reviewer_qGfb · 2025-08-07
> > > >
> > > > Is it possible to include all of these, at least in the appendix?
> > > >
> > > > Based on this, I am happy to increase my score.

---

> > > > > ### Author Response · Authors · 2025-08-07
> > > > >
> > > > > Certainly, we will include the results for the new kernels in the appendix of the final version of the paper for completeness.

---

### Decision · Program_Chairs · 2025-09-17

**Decision:**

Accept (spotlight)

**Comment:**

The reviewers unanimously recommend to accept the paper.

The only remaining criticisms are:
 - experimental results being limited to one kernel type
 - somewhat moderate novelty
Other initial criticisms have been resolved in the rebuttal and following discussions.

The remaining criticism is more than outweighed by the strong points:
 - interesting novel directions in scaling kernel methods
- contribution efficiency improvements
- strong technical merit

I congratulate the authors to their work and recommend acceptance.